# The Two Faces of Bacterial Membrane Vesicles: Pathophysiological Roles and Therapeutic Opportunities

**DOI:** 10.3390/antibiotics12061045

**Published:** 2023-06-14

**Authors:** Himadri B. Thapa, Stephan P. Ebenberger, Stefan Schild

**Affiliations:** 1Institute of Molecular Biosciences, University of Graz, Humboldtstrasse 50, 8010 Graz, Austria; 2BioTechMed Graz, 8010 Graz, Austria; 3Field of Excellence Biohealth, University of Graz, 8010 Graz, Austria

**Keywords:** membrane vesicles, blebs, biogenesis, virulence, microbial pathogenesis, antimicrobial resistance, colonization, vaccine, therapy, treatment

## Abstract

Bacterial membrane vesicles (MVs) are nanosized lipid particles secreted by lysis or blebbing mechanisms from Gram-negative and -positive bacteria. It is becoming increasingly evident that MVs can promote antimicrobial resistance but also provide versatile opportunities for therapeutic exploitation. As non-living facsimiles of parent bacteria, MVs can carry multiple bioactive molecules such as proteins, lipids, nucleic acids, and metabolites, which enable them to participate in intra- and interspecific communication. Although energetically costly, the release of MVs seems beneficial for bacterial fitness, especially for pathogens. In this review, we briefly discuss the current understanding of diverse MV biogenesis routes affecting MV cargo. We comprehensively highlight the physiological functions of MVs derived from human pathogens covering in vivo adaptation, colonization fitness, and effector delivery. Emphasis is given to recent findings suggesting a vicious cycle of MV biogenesis, pathophysiological function, and antibiotic therapy. We also summarize potential therapeutical applications, such as immunotherapy, vaccination, targeted delivery, and antimicrobial potency, including their experimental validation. This comparative overview identifies common and unique strategies for MV modification used along diverse applications. Thus, the review summarizes timely aspects of MV biology in a so far unprecedented combination ranging from beneficial function for bacterial pathogen survival to future medical applications.

## 1. Introduction

The number of multi-drug-resistant bacterial pathogens is increasing and poses a growing threat to humanity. Thus, antibiotic resistance has become a 21st-century global crisis with inefficient treatment options against resistant bacteria [1]. A recent statistical report suggests that in 2019 approximately 5 million deaths were associated with antimicrobial-resistant infections caused by bacteria, with 1.27 million directly attributable to antimicrobial resistance [2]. The “Review on Antimicrobial Resistance” published by the UK government projects that the annual death toll will be as high as 10 million by 2050 if the antimicrobial resistance problem is not addressed [3]. Apart from the human lives that are affected, the fight against drug-resistant bacteria involves an enormous financial burden for health care systems. Antimicrobials can target prokaryotic transcription and translation machineries, specific enzymes, or the cell wall. The extensive therapeutic use of antibiotics was paralleled by the emergence of diverse resistance strategies by the bacteria. Antibiotic resistance mechanisms can range from enzymatic degradation or inactivation of antimicrobial agents to alteration of the antimicrobial target sites [4]. In addition, more defensive strategies, such as extracellular matrix production or surface remodeling, can decrease the binding and uptake of antimicrobial agents [5,6]. Bacteria utilize their surface structures as protective barriers to increase resistance against antibiotics, detergents, antimicrobial peptides, and various host defense factors such as lysozymes, beta-lysin, leukocyte proteins, bile salts, and digestive enzymes [7,8,9,10,11]. In particular, the outer membrane of Gram-negative bacteria resembles an additional intrinsic permeability barrier for antimicrobial substances [12]. 

Studies on bacterial membrane vesicles (MVs) have recently added new perspectives to our understanding of antibiotic resistance and bacterial (patho-)physiology. These spherical, non-living facsimiles of their donor bacteria can be naturally secreted from the cell surface via blebbing events or originate from explosive cell lysis due to the self-assembly of membranous material [13,14,15]. Because of these diverse biogenesis mechanisms combined with different Gram statuses of bacteria, the composition and cargo of vesicles can vary, resulting in different vesicle types. Moreover, cultivation conditions and expression profiles of the donor cells can impact the composition of the released vesicles. Thus, the bacterial MV cargo covers membrane components, surface-structures, signaling molecules, nucleic acids, as well as periplasmic and cytoplasmic proteins [16,17,18,19,20,21,22,23]. The diverse chemistry of these multifactorial complexes permits various physiological activities. MVs have been demonstrated to facilitate the disposal of waste material, promote surface remodeling and nutrient acquisition, display decoys for phages and antimicrobial compounds, and act as delivery vehicles for immunomodulatory effectors, bioactive compounds, virulence factors, and nucleic acids [14,24]. MV preparations most likely consist of a heterogenous mixture of diverse vesicle types [14,15]. Depending on the cultivation condition and isolation procedure, certain MV types might be enriched or excluded. In particular, in vivo conditions and the presence of antimicrobial agents can facilitate the release of MVs with different compositions than blebbing-type MVs derived from log-phase in vitro cultures in rich media. However, we currently have neither means of purifying different vesicle types nor a defined biomarker to differentiate them. Therefore, in this review, we refer to MVs as bacterial-derived vesicles without addressing the particular vesicle type.

As highlighted in this review, exposure to antimicrobial agents can increase MV production in bacteria by triggering SOS and rpoE stress responses as well as membrane destabilization (Figure 1). Vice versa, bacterial MVs can promote resistance to antimicrobial compounds as they may act as decoys on membrane-attacking agents, contain enzymes inactivating antibiotics, promote horizontal gene transfer of resistance cassettes and enable a faster adaption via rapid surface remodeling of the donor bacterium. Notably, MVs are potent delivery vehicles for a variety of bacterial effectors affecting host cell viability and physiology. On the other hand, MVs are considered future therapeutical tools, indicated by several studies highlighting their potential in the fields of immunomodulation, vaccination, cancer therapy, and drug delivery.

To assemble the relevant studies covered in this review, a comprehensive literature search was performed in the PubMed or Google Scholar database for original, peer-reviewed research publications. No restriction on publication date was applied. Search terms comprised the term bacterial membrane vesicle in combination with keywords listed for this review, appropriate phrases found in the abstract, or titles of individual chapters. The bibliographies of relevant articles were also searched to identify other related studies.

## 2. Bacterial Membrane Vesicle Biogenesis

Since the first reports of *Escherichia coli* blebs almost 60 years ago [25], several in-depth studies have focused on the biogenesis of vesicles derived from Gram-negative and Gram-positive bacteria [13,26]. Although vesiculation is far from being understood, it is becoming increasingly evident that vesicle formation is not limited to one particular mechanism. Depending on the bacterial species, growth status, and environmental conditions, MVs can arise in multiple ways that may even occur simultaneously in a bacterial population. Notably, several models of vesicle release are connected to signals or triggers that bacteria face during infection or upon antimicrobial treatment.

In Gram-negative bacteria, changes in the surface composition can initiate bulging events resulting in blebbing of the outer membrane and MV formation. For example, the Pseudomonas quinolone signal (PQS), a quorum-sensing signaling molecule of *Pseudomonas aeruginosa*, can interact with the lipid A moiety of the LPS and generate repulsive forces triggering membrane curvature and vesicle release [27]. Apart from MV formation, PQS has been linked to virulence and host colonization of *P. aeruginosa* by several studies. PQS promotes biofilm formation, virulence factor expression, cytotoxicity, and modulation of host immune responses [28]. Being a potent iron chelator, PQS is also involved in iron acquisition, promoting the growth of *P. aeruginosa* in iron-limiting conditions [29]. The iron limitation is frequently faced by bacteria during colonization of the human host, as our body stores iron bound to various proteins, such as hemoglobin, haptoglobin, transferrin, ferritin, and lactoferrin [30]. Iron-limiting conditions also promote another reported mechanism of vesicle blebbing. The “maintenance of lipid asymmetry” (Mla)-pathway, also known as VacJ/Yrb-system, comprises a widely conserved bacterial phospholipid transporter system of Gram-negative bacteria shuffling phospholipids between the outer and inner membrane [31]. Inactivation or transcriptional silencing of the Mla transporter results in an accumulation of phospholipids in the outer membrane and increased vesiculation in a variety of bacteria, such as *E. coli*, *Vibrio cholerae,* and *Haemophilus influenzae* [32]. Excessive amounts of phospholipids likely induce bulging of the outer membrane resulting in vesicle secretion. Importantly, iron limitation results in transcriptional silencing of the mla operon, which may also contribute to the observed repression of the system during host colonization, facilitating vesicle release in vivo [32,33]. The limitation of nutrients other than iron has also been reported to trigger vesiculation via outer membrane phospholipid accumulation. In *Neisseria meningitidis*, sulfate depletion activates phospholipid biosynthesis driving excessive production, which finally results in their accumulation in the outer membrane and increased vesiculation [34].

LPS modulation can also trigger bacteria vesicle release. For example, MVs derived from *P. aeruginosa* and *Porphyromonas gingivalis* are enriched for negatively charged LPS molecules, which may cause local destabilization of the outer membrane integrity by an accumulation of negative charges [35,36]. In *Salmonella typhimurium*, deacylation of the lipid A moiety of the LPS was shown to induce vesiculation as layers of LPS with underacylated lipid A likely change into a cone-shape structure resulting in a positive curvature of the outer membrane [20,37]. Expression of the responsible deacylase PagL is induced in *S. typhimurium* upon internalization by macrophages suggesting increased vesiculation of the pathogen in host cells [38,39]. 

Increased vesiculation upon destabilization of the bacterial cell surface is confirmed by several studies. In Gram-negative bacteria, mutations reducing covalent linkages of the outer membrane to the peptidoglycan in the periplasm generally result in hypervesiculation. Noteworthy examples are loss-of-function mutants of Braun’s lipoprotein Lpp, the outer membrane porin OmpA and the Tol-Pal complex, which have been reported to exhibit increased vesiculation in diverse bacterial species [40,41,42,43,44,45,46,47,48]. Although such mutational phenotypes might not be considered naturally relevant, they highlight the potential of vesicle release triggered by stressors lowering the integrity of the bacterial envelope. Furthermore, in *E. coli*, *V. cholerae*, and *S. typhimurium,* Lpp and OmpA levels are controlled by regulatory sRNAs, such as VrrA, MicA, and Reg26 [49,50,51]. These regulatory circuits also provide a link between MV formation and envelope stress response as all three sRNAs are positively controlled by RpoE, an alternative sigma factor activated during exocytoplasmic and membrane stress [49,50,52]. Several studies demonstrate that impairment of the RpoE pathway and disruption of periplasmic equilibrium, e.g., by the accumulation of misfolded proteins or peptidoglycan breakdown products, results in increased MV release [50,53,54,55,56]. Indeed, misfolded proteins or peptidoglycan breakdown products can be found in the cargo of MVs upon stress exposure [54,55,57]. Thus, vesiculation can be seen as a stress-induced response enabling bacteria to maintain envelope homeostasis via the release of misfolded products or osmolytes, which may otherwise elevate turgor pressure and interfere with cell physiology. 

Similar to Gram-negative bacteria, destabilization of the cell envelope has also been reported as a trigger for MV secretion in Gram-positive bacteria. For example, in *Staphylococcus aureus* amphipathic, alpha-helical, surfactant-like modulins, and cell-wall hydrolases, i.e., autolysins, disrupt the cytoplasmic membrane and weaken peptidoglycan linkages to promote MV release [58,59]. Similarly, a prophage-encoded endolysin creates holes in the thick peptidoglycan layer of *Bacillus subtilis*, allowing cytoplasmic membrane material to protrude into the extracellular milieu and eventually punch off as MV [60]. Degradation of peptidoglycan by phage-encoded endolysins can also result in vesicle production in Gram-negative bacteria [61,62,63,64]. Although donor bacteria do not survive this process, also known as explosive cell lysis, the remaining bacterial population may benefit from such MVs formed by self-assembly from membranous material liberated by lysed cells (for details, see below). Explosive-type vesicles may contain substantial amounts of the inner membrane as well as cytoplasmic components, which are otherwise rather excluded from blebbing-type MVs in Gram-negative bacteria [14]. Notably, several antimicrobial and genotoxic agents can stimulate endolysin expression of prophages and explosive cell lysis via induction of the bacterial SOS response, representing a conserved DNA repair mechanism activated by DNA damage in bacteria. 

In line with the aforementioned MV biogenesis mechanisms, it is not surprising that antimicrobial therapy likely increases MV formation by inducing envelope and SOS stress responses as well as damaging or destabilizing the bacterial cell surface. Increased MV production in *P. aeruginosa* upon gentamicin exposure was already reported 30 years ago [36,65,66]. In addition to virulence factors, gentamicin-induced vesicles contain substantial amounts of DNA, already suggesting the hypothesis that MVs could act as DNA delivery vehicles and play an important role in horizontal gene transfer [36]. Aside from inhibition of protein synthesis, the authors hypothesized that cationic gentamicin could also destabilize bacterial membranes by interaction with anionic LPS in the outer membrane [66]. Similarly, polymyxin B and colistin can destabilize electrostatic interactions between LPS molecules by replacing bivalent cations, such as Mg^2+^ and Ca^2+^, which increases MV release in a variety of bacteria [56,67,68,69,70,71]. In addition, host-derived antimicrobial factors targeting membrane stability, such as cationic antimicrobial peptides or bile salts, have been reported to promote MV secretion [72,73]. 

As gentamycin and polymyxin B have been reported to induce SOS response and pro-phage activation in enterohemorrhagic *E. coli* (EHEC) and *Acinetobacter baumannii*, explosive cell lysis may also contribute to increased MV production for several of the above mentioned antimicrobial compounds [69,74]. In addition, fluoroquinolones, i.e., ciprofloxacin, and genotoxins, i.e., mitomycin C, trigger SOS responses in diverse Gram-negative and Gram-positive bacterial species, which results in increased MV formation [61,74,75]. Finally, β-lactam antibiotics targeting the peptidoglycan synthesis also increase MV production by weakening the cell wall, as demonstrated for several bacterial pathogens, e.g., *P. aeruginosa*, *A. baumannii*, EHEC, *Stenotrophomonas maltophilia*, *S. aureus*, and *Streptococcus pyogenes* [69,74,75,76,77,78]. Of major concern are studies indicating that MVs released upon antibiotic exposure can be enriched for virulence factors and exhibit higher cytotoxicity. For example, Shiga toxin can be enriched in EHEC MVs upon the presence of antibiotics inducing the SOS response and pro-phage activation resulting in higher expression of the phage-encoded Shiga toxin [74]. These observations reinforce the recommendations to avoid the treatment of EHEC infections with certain antibiotics, as this increases toxin production and the risk for hemolytic urinary syndrome [79]. Proteins involved in antibiotic resistance as well as virulence factors are enriched in *Elizabethkingia anophelis* MVs grown in the presence of antibiotics [80]. MVs derived from *P. aeruginosa* in the presence of SOS response-inducing ciprofloxacin contain virulence-associated proteins, such as the M48 family peptidase, cytochrome c and OprG, which might explain their higher cytotoxicity in macrophages than MVs from *P. aeruginosa* grown without antibiotics [64].

## 3. Physiological Function of MVs within the Context of Human Health and Disease

Bacteria tremendously invest in diverse synthesis, transportation, and secretion pathways to build and maintain their membranes and surface structures. The Liberation of MVs seems paradoxical and energetically costly [24]. Nonetheless, to our knowledge, all bacterial cultures analyzed so far contain MVs, and non-vesiculating mutants have not been reported. Thus, MVs likely have beneficial functions for their donor bacteria or, in a broader sense, for the bacterial population. In particular, vesiculation of bacterial pathogens seems to facilitate colonization fitness, counteract antimicrobial stressors, and mediate effector delivery to the host. Herein, we highlight some proposed roles of MVs in promoting virulence and pathogenicity.

### 3.1. Impact of MVs on In Vivo Adaptation and Fitness

MVs generally represent a similar membrane structure and composition as the donor cells. Thus, agents directed against bacterial surface structures may also bind to MVs, thereby lowering the effective dose targeting living bacterial cells and facilitating bacterial survival. Indeed, several reports demonstrate that MVs can act as decoys against antibodies, bacteriophages, or antimicrobial factors. 

MVs can inhibit bacteriophage infection, as reported for virulent phages ICP1, ICP2, and ICP3 of *V. cholerae* and the bacteriophage T4 of *E. coli* [67,81]. Thus, MVs should be considered as phage-neutralizing factors along phage therapy as MVs likely affect the efficacy of such intervention strategies. *Moraxella catarrhalis* MVs carrying the immunoglobulin (Ig) D-binding superantigen MID can redirect B cell responses towards the production of polyclonal IgM, which are not directed against the pathogen of the respiratory tract [82]. Thus, *M. catarrhalis* uses MVs as decoys to redirect the adaptive humoral immune response. 

Mechanisms to counteract host-derived antimicrobial peptides are a key feature for several bacterial pathogens. On the one hand, sublethal concentrations of these stressors trigger MV formation; on the other, MVs can protect bacteria against a variety of antimicrobial peptides. For example, *E. coli* MVs act as a decoy and carrier vehicle for PMAP-36, CATH-2, and LL-37 [83]. In similar studies, MVs enhanced the survival of enterotoxigenic *E. coli* (ETEC) in the presence of polymyxin B and colistin as well as *Helicobacter pylori* against LL-37 [67,84]. MVs from *A. baumannii* function as efficient decoys for polymyxin B, ensuring survival and virulence of the pathogen in the *Galleria mellonella*-infection model even upon exposure to the antimicrobial agents [85]. MVs from *S. aureus* protect the bacteria from membrane-targeting antibiotic daptomycin in vitro and ex vivo in whole blood [75]. MVs from *B. subtilis* have been shown to be protective against environmental stressors such as surfactants, O_2_ depletion, starvation, and thermal shock [86,87]. Increased vesiculation of *H. influenzae* and *Salmonella enterica* or the addition of MVs to cultures of *Aggregatibacter actinomycetemcomitans* confers serum resistance and survival in the presence of an active complement [32,88,89]. In the case of *S. enterica*, MVs not only bind membrane-attacking components but can also carry enzymes, i.e., PagC, actively degrading the C3 component of complement in complement-mediate killing [89]. The expression of PagC, as well as the upregulation of MV release, depends on the activation of the two-component-system PhoPQ, which is also required for intracellular survival in host cells [90]. 

Activation of the two-component systems PhoPQ and PmrAB by low bivalent cation concentrations, acidic pH, toxic metals, and cationic antimicrobial peptides not only enhance vesiculation by destabilizing the outer membrane but also induce LPS modulations pivotal for *Salmonella’s* intracellular survival and replication. A comprehensive study analyzing the outer membrane and MV composition over time upon shifting bacteria to PhoPQ- and PmrAB-inducing conditions shows that modified LPS remain in the bacterial membrane, whereas the unmodified negatively-charged LPS is rather removed from the surface via MVs. The facultative pathogen *V. cholerae* uses a similar strategy to rapidly change its surface. Upon oral ingestion by the host, *V. cholerae* induces MV release to liberate unfavorable surface compounds from its surface [33]. Vice versa, this facilitates the accumulation of glycine-modified LPS and depletion of the outer membrane porin OmpT in the outer membrane. These surface changes are critical to achieving full colonization fitness and resistance to host-derived factors such as antimicrobial peptides and bile. Thus, MV release can promote surface remodeling upon changing environments along the bacterial lifecycles, such as host entry by facultative pathogens.

It should be noted that MV protection is not limited to the donor bacterium or species but rather can serve the entire bacterial population. MVs carrying degradative enzymes or antibiotic resistance cassettes can facilitate antimicrobial resistance (Figure 2). For example, MVs from *E. coli* not only protected *E. coli* from membrane-active antibiotics such as colistin and melittin but also *P. aeruginosa* and *Acinetobacter radiodioresistens* [91]. Proteomic analyses revealed that the MVs carry proteases and peptidases involved in the degradation of these antibiotics. These results also explain that MVs did not protect any of these bacteria against other antibiotics, such as ciprofloxacin, streptomycin, and trimethoprim. Indeed, the impact of MVs on short-term antibiotic resistance can be explained by antibiotic-degrading enzymes, such as β-lactamases, in their cargo [92]. Early reports already highlight the packaging of β-lactamases in MVs of β-lactam resistant *P. aeruginosa* isolated from cystic fibrosis patients [77]. Release of such MVs at the infection site likely reduces antibiotic efficacy. Another study highlights the cross-protectivity of *E. coli* MVs carrying the New Delhi metallo-β-lactamase (NDM-1). These NDM-1-containing MVs facilitated the survival of carbapenem-susceptible strains of *E. coli* and *P. aeruginosa* upon treatment with meropenem in a live infection model [93]. 

In addition to carrying enzymes for the degradation of antibiotics that confer temporary protection, MVs can also promote the horizontal gene transfer of antibiotic resistance genes and thus favor the emergence of multidrug-resistant strains. MVs from carbapenem-resistant clinical strains of *A. baumannii* harbored the plasmid-borne blaOXA-24 gene, which could be transferred to carbapenem-susceptible *A. baumannii* isolates [94]. Similar observations of MV-mediated delivery of plasmid-encoded resistance genes have been reported for other ESKAPE pathogens, such as *Klebsiella pneumoniae* and *P. aeruginosa* [95,96]. Notably, biofilm-derived MVs of *P. aeruginosa* showed higher efficiency in horizontal gene transfer than MVs isolated from planktonic cultures [96]. This might increase the likelihood of such events in vivo, where *P. aeruginosa* forms biofilms in the lung. In addition to antibiotic resistance genes, MVs can also act as delivery vehicles for virulence genes. For example, MVs from *E. coli* O157:H7 can transfer virulence genes, such as *eae*, *uidA*, *stx1*, and *stx2*, to other *E. coli* or *S. enterica* [97,98]. Horizontal gene transfer has also been shown for *P. gingivalis* MVs, transferring virulence and erythromycin resistance genes to sensitive strains of *P. gingivalis* [99]. Recent advances along the development of improved high-resolution imaging technologies might be promising approaches to allow the detection and identification of MVs harboring resistance markers [100]. Moreover, therapeutic agents reducing or blocking MV release might be available in the future and could be valuable tools to counteract such MV-based antimicrobial resistance mechanisms [93]. 

### 3.2. Effector Delivery by MVs

MVs derived from bacterial pathogens have been, in particular, studied along with their role as delivery vehicles of toxins and virulence factors. Indeed, a variety of effectors modulating host cell fitness or physiology have been found in the MV cargo. The small size, their efficient internalization by host cells, as well as their potential for systemic distribution, may allow the delivery of effectors to distant sites in concentrated doses [14,101]. Using lipophilic dyes, transmigration across the intestinal epithelium was observed for *Bacteroides thetaiotaomicron* MVs after oral administration, and systemic distribution was observed within hours into various tissues, with the highest accumulation in the liver [102]. Meningococcal vesicles could be detected in the cerebrospinal fluid of infants infected with *N. meningitidis* as well as in sera of patients suffering from multiple organ failure after infection for *N. meningitidis* serogroup B [103,104].

Several MVs derived from bacterial pathogens can fuel pro-inflammatory responses. Exposure of human umbilical vein endothelial cells with MVs from pathogenic *E. coli* activates the pro-inflammatory NF-κB pathway as well as coagulation cascades, thereby contributing to sepsis symptoms [105,106]. Intraperitoneal injection of *E. coli* MVs in the murine model induced a systemic, pro-inflammatory immune response closely resembling sepsis [107]. The intratracheal challenge with MVs from *K. pneumoniae* in a neutropenic mouse model increases cytokine expression levels and induces a severe lung pathology similar to the one observed when colonized with the pathogen itself [108]. 

MVs from *H. pylori* induce pro-inflammatory interleukin (IL)-8 responses in stomach tissue and might thereby contribute to low-grade gastritis typical for *H. pylori* infection [109]. *H. pylori* MVs carry a cocktail of host cell modulating factors, including the cytotoxin-associated gene A CagA, the vacuolating cytotoxin VacA, adhesins SabA and BapA, as well as outer inflammatory protein OipA [110]. The presence of VacA on *H. pylori* MVs not only enhances binding to gastric epithelial cells but also broadens internalization routes into host cells, while VacA-depleted MVs are predominantly taken up by clathrin-mediated endocytosis [111]. Thus, uptake routes of *H. pylori* MVs may vary depending on their cargo. 

The MV-associated porin PorB from *Neisseria gonorrhoeae* was shown to target mitochondrial membranes resulting in loss of mitochondrial integrity and activation of apoptosis [112]. Similarly, *A. baumannii* MVs contain OmpA, which induces mitochondrial fragmentation and cytotoxicity [113]. *Bacillus anthracis* MVs are cytotoxic to macrophages due to their cargo comprising various toxins of the pathogen, such as the protective antigen, lethal factor, edema toxin, and anthrolysin [114].

Proteolytic activity of gingipains present in MVs derived from the periodontal pathogen *P. gingivalis* was correlated with tissue disruption and loosening cell–cell connections, which promotes MVs penetration into deeper tissues and eventually even crossing of the blood–brain barrier, causing inflammatory responses [115,116,117]. The cytolethal distending toxin (Cdt) from *Campylobacter jejuni*, the zinc-dependent non-lethal metalloprotease toxin from *Bacteroides fragilis*, as well as the cytotoxic necrotizing factor type 1 from uropathogenic *E. coli* (UPEC), were reported to be associated with MVs and effectively delivered to intestinal epithelial cells via MVs driving pathological symptoms of the respective disease [118,119,120]. 

ETEC MVs contain several virulence factors, such as the heat-labile enterotoxin (LT), EtpA, CexE, TibA, and flagellins [121,122,123]. More than 95% of the LT, responsible for secretory diarrheal symptoms, is associated with MVs due to its affinity to the core region of the LPS [124,125]. The binding of LT to GM1 gangliosides in cholesterol-rich membrane areas of the host cells also seems responsible for the MV uptake via lipid rafts [126]. LT has 80% homology to the cholera toxin (CT) from *V. cholerae*, the causative agent of cholera disease. Although, in this case, the majority of enterotoxin is released via the type-2-secretion system into the extracellular milieu, about 15% of the CT was found to be associated with MVs [21,127]. In contrast to type-2-secreted CT, the uptake of MV-associated CT is not GM1 ganglioside dependent but rather utilizes caveolin-mediated endocytosis [21,128]. In addition to broadening the internalization routes, CT associated with MVs is fairly protected against degradation by intestinal proteases, which extends the half-life of the toxin in the intestinal environment [21]. This might indicate luminal localization of the CT in *V. cholerae*, whereas LT in ETEC MVs is likely to surface exposed to enable GM1-binding. 

Differential localization of MV-associated virulence factors has also been reported for EHEC [129,130]. Comprehensive microscopical evaluation combined with proteinase K-sensitivity assays of MVs indicates that the Shiga toxin 2a and Cdt are located inside the vesicles, while hemolysin and flagellin were found on the vesicle surface [129]. Dynamin-dependent endocytosis of MVs derived from EHEC O104:H4 by intestinal cells induces a caspase 9-mediated apoptosis and pro-inflammatory cytokine secretion [130]. In a more detailed study, the same group demonstrates a complex sequential separation of effectors associated with EHEC 0157 MVs during intracellular trafficking in human epithelial and endothelial cells [129]. Shiga toxin 2a and the B-subunit of Cdt are released from the MVs in early endosomes. The catalytically active A1-fragment of the Shiga toxin eventually reaches the cytosol after trafficking through the Golgi apparatus and the endoplasmic reticulum. The B-subunit of Cdt is translocated to the nucleus after its retrograde transport to the endoplasmic reticulum, where it causes DNA damage. Other effectors, such as the A- and C-subunits of the Cdt as well as the hemolysin, remain associated with the MVs and end up in the lysosomes. There, the hemolysin detaches to target mitochondria.

Membrane vesicles of several bacteria have been reported to carry RNA as their cargo, which may modulate host cell physiology in addition to immunostimulation by TLR-7, -8, or -9 recognition [14,131,132,133]. The small RNA sRNA52320 in *P. aeruginosa* MVs downregulates the expression of genes involved in the MAPK pathway in human airways epithelial cells, which suppresses the LPS-induced IL-8 response as well as CXCL1 secretion and neutrophil infiltration in a mouse lung [134]. Similarly, two sRNAs present in *H. pylori* MVs also attenuate the IL-8 response in human gastric adenocarcinoma cells [135]. *Legionella pneumophila* uses MVs to translocate bacterial sRNAs into host cells reducing expression of RIG-I, IRAK1, and cRel, which results in the downregulation of IFN-β. Interestingly, the identified sRNAs exhibit a dual function as these trans-kingdom sRNAs also regulate life cycle adaptation and protein synthesis in *L. pneumophila* [136]. *Streptococcus sanguinis* and periodontitis-related bacteria, such as *A. actinomycetemcomitans, P. gingivalis,* and *Treponema denticola* have also been reported to secrete sRNA via MVs, which mediate suppression of several cytokines in eukaryotic cells [137,138]. Similar to proteins, the RNA content of MVs seems protected from degradation allowing long-distance delivery. For example, intracardiac injection of MVs from *A. actinomycetemcomitans* can cross the blood–brain barrier and deliver the RNA cargo to the mouse brain, enhancing TNF-α production [139]. 

## 4. Potential Therapeutic Applications of Bacterial Membrane Vesicles

As summarized in the chapter above, bacterial MVs can contribute to the in vivo fitness of pathogens and protect them from antimicrobial treatment in a variety of ways. While we must acknowledge the efficacy of these strategies, the exploitation of fundamental features of MVs in conjunction with the genetic engineering of donor strains that secrete MVs with improved properties for therapeutic applications has become a focus of several research initiatives (Table 1).

MVs are packed with microbe-associated molecular patterns (MAMPS), such as LPS, flagella, (lipo)-teichoic acids, peptidoglycan, and nucleic acids, which are recognized by pattern recognition receptors (PRRs) of host cells triggering inflammatory responses [131]. Moreover, MVs can be effectively internalized by diverse host cells, including non-phagocytic epithelial and endothelial cells, utilizing various uptake pathways such as clathrin- and caveolin-mediated endocytosis, lipid raft-mediated uptake, and micropinocytosis [101]. Thus, aside from extracellular recognition mainly mediated via TLRs, MV content can also be recognized by cytosolic receptors of the host cells, such as the NOD1, NOD2, caspases, and guanylate-binding proteins involved in the NLRP3 inflammasome activation [14,131]. MVs can directly interact with immune cells and modulate B and T cell responses [140]. MVs from *N. meningitidis*, *H. pylori*, *M. catarrhalis*, and *H. influenzae* have been reported to interfere in B and T cell responses, facilitating bacterial survival and colonization [82,141,142,143]. This potent immunomodulatory activity of MVs might be exploited for future therapeutical applications. Among therapeutical applications of MVs, their use as vaccine candidates is by far best studied and has already resulted in licensed vaccines [144,145,146,147]. As non-living facsimiles of their donor cells, MVs present surface-exposed antigens in their native conformation. Vaccine candidates based on bacterial MVs have proven highly immunogenic, resulting in robust, protective immune responses. Genetically modified MVs not only increased yield but also improved the safety profile, e.g., by reducing LPS endotoxicity. Especially in the age of emerging antibiotic (multi)-resistance, alternative therapies and measures to prevent bacterial infections should be reinforced. Therefore, MVs should be considered promising alternative vaccine candidates [148,149].

### 4.1. Modulation of (Innate) Immune Responses

The immunomodulatory potential of bacterial MVs can facilitate colonization fitness and virulence of pathogens, but several reports also highlight beneficial properties of MVs from probiotic strains, such as *E. coli* Nissle, *B. fragilis,* and *Lactobacillus* ssp. However, to our knowledge, currently, none of the below-mentioned findings are comprehensively tested in clinical trials. 

MVs derived from *E. coli* Nissle can stimulate proliferation, immune-related enzymatic activities, phagocytic function, and anti-inflammatory cytokine release in macrophages [150]. Immunomodulatory activity of *E. coli* Nissle MVs seems versatile as they have also been shown to induce diverse pro- and anti-inflammatory cytokine responses in peripheral blood mononuclear cells and monocyte-derived dendritic cells (i.e., IL-6, IL-8, TNF-α, and anti-inflammatory IL-10) as well as in human intestinal epithelial cells via the cytosolic NOD1 receptor after internalization (i.e., IL-6 and IL-8) [150,151,152]. MVs from *E. coli* Nissle can also improve and restore epithelial barrier function by modulating protein involved in tight junctions in intestinal epithelial cells [153,154]. Oral treatment of DSS-induced colitis in mice with *E. coli* Nissle MVs reduced pro-inflammatory cytokine levels, tissue damage, and weight loss, confirming that the beneficial properties of these MVs are also applicable in systemic in vivo models [155]. 

MVs derived from the commensal *B. fragilis* carry polysaccharide A (PSA), which promotes the development of IL-10-producing regulatory T cells and thereby suppresses inflammation in animal models of inflammatory bowel disease [156,157]. Oral administration of PSA-containing *B. fragilis* MVs successfully prevents experimental TNBS-induced colitis in mice, which requires TLR-2-dependant sensing of PSA by dendritic cells, promoting anti-inflammatory cytokine release and regulatory T cell production [157].

MVs derived from *Lactobacillus paracasei*, *Lactobacillus plantarum* Q7, and *Lactobacillus rhamnosus* have also been reported to confer protection against DSS-induced colitis in the mouse model [158,159,160]. Although the exact mechanisms remain to be elucidated, a recent study indicates that tryptophan metabolites being a cargo of *L. rhamnosus* MVs, could mediate some of the beneficial effects [161]. Engineering of MVs might also further improve their properties, as highlighted for fucoxanthin-loaded MVs from *L. plantarum* significantly suppressing pro-inflammatory cytokine expression in mice suffering from DSS-induced colitis [162].

### 4.2. MV-Based Vaccine Candidates

The first successfully licensed MV-based vaccines, known as MenBvac^®^, MeNZB^®,^ and VA-MENGOC-BC^®^, were effectively used against epidemic outbreaks of group B meningococcus in Norway, Chile, Cuba, and New Zealand [144,145,146,147]. To reduce toxicity, LPS was removed from *N. meningitidis* MVs by detergent extraction, leaving the highly variable PorA protein as the immunodominant antigen [163]. A multicomponent meningococcal B MV-based vaccine (4CMenB, Bexsero^®^) was approved by the European Commission in 2013, which comprises a blend of *N. meningitidis* MVs mixed with three highly immunogenic antigens (fHbp, NadA, and NHBA) identified by reverse vaccinology [164]. Studies show that the vaccine also offers some degree of protection against gonorrhea, probably due to the homology of the two closely related species, i.e., *N. meningitidis* and *N. gonorrhoeae* [165,166]. In summary, these *Neisseria* vaccines represent the first globally approved, commercially available MV-based vaccines and have proven decent efficacy and safety, which might pave the way for future vaccine candidates relying on MV technology [167].

Several studies successfully characterized MVs derived from various bacterial pathogens as vaccine candidates (reviewed elsewhere: [149,168,169]). MVs derived from *A. baumannii* induced robust antibody titers protecting mice from pneumonia and sepsis in challenge studies using multidrug-resistant clinical isolates [170,171]. Vaccination with MVs from *S. aureus* elicits humoral and cellular immune responses, conferring protection against challenges with lethal doses of *S. aureus* [172]. Thus, MVs could be promising vaccine candidates to fight infections caused by ESKAPE pathogens. Aside from *S. aureus*, immunization with MVs from other Gram-positive species, such as *Streptococcus pneumoniae* and *Mycobacterium tuberculosis*, shows high immunogenicity in mice and protects against challenges in animal models [173,174,175]. Especially in the elderly, pneumonia cases caused by pneumococcal infections are constantly increasing, and the emergence of antibiotic resistance limits therapeutic interventions [176]. Vaccination is considered the most promising approach, but the efficacy of the currently licensed vaccines against community-acquired pneumonia is controversially discussed [177,178,179]. As for many other infectious diseases, alternative vaccine candidates are desperately needed. Thus, a recent report describing a spray-dried formulation of pneumococcal MV-loaded vaccine microparticles are promising approaches, which require further investigation with regard to animal testing [180]. 

MVs from Gram-negative *Bordetella pertussis,* the causative agent of whooping cough, induced a balanced Th1/Th17 and Th2 response, which is considered beneficial for protection [181,182]. In addition, MV immunization showed higher IgG levels, lower systemic pro-inflammatory cytokine responses, and enhanced splenic gene expression compared to the classical whole-cell pertussis vaccine [181,182]. Proteomic analyses revealed quite a complex composition of *B. pertussis* MVs, providing some explanation for the broad and balanced humoral response induced upon immunization, with Vag8, BrkA, and LPS being the most immunogenic antigens [183]. A diverse composition of non-typeable *H. influenzae* (NTHi) MVs might also explain the broad immune response induced upon vaccination, ensuring cross-protection against several diverse NTHi strains [184,185]. Thus, MV-based vaccines could overcome the genetic heterogeneity and surface-exposed antigenic variability of NTHi strains being the key limitations for vaccine development against these pathogens of the respiratory tract [186,187]. Similarly, mixtures of MVs derived from different *Shigella* or *V. cholerae* strains induced broad-spectrum antibody responses against the LPS and surface proteins of respective pathogens, conferring protection against all serogroups causing shigellosis or cholera [188,189]. Immunization with a mixture of MVs derived from *V. cholerae* and ETEC induced a protective immune response against both intestinal pathogens, highlighting the potential to develop a combined MV-based vaccine [190]. 

As highlighted above, bacterial MVs may contain virulence factors mediating cytotoxicity in host cells. In addition, MVs from Gram-negative bacteria carry substantial amounts of LPS harboring the lipid A endotoxin. Thus, MV immunization, especially upon invasive route administration, could be associated with adverse effects such as decreased cell viability and exacerbated inflammatory reactogenicity. However, MVs harboring under-acylated LPS without depleted virulence factors or toxins produced by genetically modified donor strains have been reported for several pathogens such as *N. meningitidis*, *Shigella sonnei*, *V. cholerae,* and ETEC [190,191,192]. Animal studies and human clinical trials using detoxified MVs are promising as they demonstrate continued immunogenicity and protective efficacy [190,193,194,195]. One of the most advanced MV-based vaccine candidates is 1790GAHB, conferring protection against *S. sonnei*. Phase I and II clinical trials performed in Europe and Kenya elicited a strong immune response with bactericidal activity in immunized volunteers, significantly increased by a booster vaccination [192,195,196,197,198]. Understanding of biogenesis mechanisms also allowed the construction of hypervesiculating donor strains increasing the yield [199,200]. Thus, MVs from genetically modified donor strains could be promising candidates for low- and middle-income countries based on relatively low production costs, high purity, and yield, as well as the robust efficacy of the vaccine candidates [201]. Although the MV-based vaccine showed satisfactory tolerability in clinical trials, the risk of systemic reactogenicity remains to be carefully monitored and evaluated [201]. In particular, potentially combinatory effects of MVs, absorbent agents, such as Alhydrogel, adjuvants, and aluminum hydroxide, as well as differential reactogenicity in children and adults, should be taken into consideration [194,202,203,204]. 

Genetic engineering of MV donor strains also offers the possibility to increase yield and incorporate new antigens of interest. Diverse decoration strategies to display heterologous polysaccharide and protein antigens on MVs have been successfully established over recent years. For example, a hypervesiculating *tolRA*-knock-out strain of *E. coli* was glycoengineered to produce MVs decorated with the O-antigen polysaccharide of *Francisella tularensis*, the causative agent of tularemia [205]. Upon immunization in mice, the glycoconjugated MVs elicited a significant antibody response directed against the *F. tularensis* O-antigen and prolonged survival of immunized mice upon challenge with the pathogen [205]. MVs from *E. coli* genetically glycoengineered to display K1 and K2 capsule polysaccharides of hypervirulent *K. pneumoniae* were also found to be immunogenic and efficacious, protecting mice against lethal infection with the ESKAPE pathogen [206]. 

Genetically modified MVs can also simultaneously serve as adjuvant and carrier platforms of antigens with low immunogenicity. This is illustrated by a proof-of-principle study expressing GFP fusions to the pore-forming hemolytic protein ClyA, which ensures localization on *E. coli* MVs. Comparative studies in the murine model showed a robust anti-GFP response upon immunization with ClyA-GFP-loaded MVs, whereas immunization with GFP alone elicited significantly lower titers [207]. The ClyA fusion technology was also used to generate *E. coli* MVs carrying Omp22, an outer membrane protein of *A. baumannii*. Immunization of mice with Omp22-decorated MVs resulted in high anti-Omp22 immunoglobulin titers and protection against a clinically lethal *A. baumannii* isolate [208]. 

Bacterial autotransporter platforms, such as Hbp and AIDA, have been proven to be efficient methods for presenting antigens on the surface of MVs. In detail, Hbp fusions were successfully used to present up to three *M. tuberculosis* antigens on *E. coli* MVs or the *Clamydia trachomatis* major outer membrane protein on *S. enterica* MVs [209]. Fusion to the transporter domain of the *E. coli* AIDA adhesin allowed decoration of *S. enterica* MVs with two Leishmania antigens, which elicited specific antibody responses against the parasite antigens in mice upon subcutaneous immunization [210].

The SpyTag/SpyCatcher system offers a broader approach to decorating bacterial MVs with heterologous proteinaceous antigens [211]. Along MV vaccine design, the SpyTag or the SpyCatcher domains can be fused to proteins, such as membrane vesicle anchors and antigens of interest, to enable interaction and covalent linkage formation between the two protein candidates. This click strategy was used to covalently link SpyTag-fused *S. aureus* antigens to OmpA-SpyCatcher fusions located on *E. coli* MVs. The “click” MV vaccine candidate induced strong humoral and cellular immune responses in the murine model conferring protection against *S. aureus* lethal challenge [212].

The SARS-CoV-2 pandemic has taken the heterologous antigen display strategies of MV-based vaccine design to a new level. It has become evident that fast development of effective, long-lasting, easily administrable vaccines offering sufficient production capacity and simple storage conditions are required to fight such explosive unpredictable outbreaks. The ClyA fusion strategy combined with a high-pressure homogenization technology was used to generate *E. coli* MVs decorated with substantial amounts of the receptor binding domain (RBD) of the SARS-CoV-2 virus [213]. Efficacy was evaluated by subcutaneous immunization studies in mice, which elicited SARS-CoV-2-specific humoral and cellular immune responses. Detoxified MV donor strains of *V. cholerae* and ETEC have not only been further genetically modified to increase MV yield but also display the SARS-CoV-2 RBD antigen on their surface [200]. To achieve the latter, a fusion technology based on the major outer membrane lipoprotein Lpp and the outer membrane porin OmpA was used. The signal sequence and N-terminal acylation site of the Lpp fragment direct the fusion protein to the outer membrane, while the OmpA portion with membrane-spanning segments allows stable integration into the outer membrane with the C-terminally fused antigen located externally. Mucosal immunization with the RBD-decorated MVs using the intranasal administration route induced robust immune responses against both the MV donor pathogens and the SARS-CoV-2 spike protein. Another potential intranasal MV-based vaccine candidate is based on a recombinant, stabilized version of the spike protein linked to the LPS-binding peptide sequence mCramp of *N. meningitidis* [214]. Intranasal administration of the spike protein-decorated, detoxified *N. meningitidis* MVs induced protective immune responses in mice and hamsters with high IgG and IgA levels in the nose and lungs. In light of desperately needed vaccines, the adjuvant properties of MVs have been exploited in simple approaches mixing MVs with viral proteins. A formulation of *N. meningitidis* MVs, similar to the approved MV-based *N. meningitidis* vaccine Bexsero^®^, combined with recombinant RBD, has also been reported to stimulate humoral immune responses against the spike protein [215]. The versatile SpyTag/SpyCatcher system was also successfully used to decorate *S. typhimurium* MVs from a hypervesiculating, detoxified strain carrying Hbp-SpyCatcher fusions with Spytag-RBD fusion proteins. Intranasal immunization with these RBD-loaded MVs resulted in neutralizing antibody responses conferring protection against the SARS-CoV-2 challenge in a hamster model [216]. 

In addition to SARS-CoV-2, bacterial MVs have been described as antigen display systems for other viruses. For example, the expression of a fusion protein composed of the RBD from the MERS-CoV and the hemagglutinin (HA) from a H1N1 influenza virus strain in *E. coli* resulted in MVs carrying this chimeric antigen. Immunization studies in mice demonstrated the rise of neutralizing Abs titers against both viruses and protection against challenges with the influenza virus [217]. Expression of envelope E protein domain III from different serotypes of the dengue virus in *S. aureus* yielded a multivalent MV vaccine candidate, which induced an antibody response against all dengue virus serotypes in the murine model [218]. MVs derived from *Lactococcus lactis* displaying the hepatitis C virus core antigen and the recombinant fusion protein consisting of a truncated core and NS3 combined with MVs from *N. meningitidis* have been suggested as MV-based vaccine candidates against hepatitis C virus infections, which can lead to liver failure and cancer [219,220]. 

Some of the described antigen display systems have also been used to decorate bacterial MVs with tumor antigens. The efficient presentation to antigen-presenting cells and self-adjuvant properties of MVs could at least partially overcome the low immunogenicity observed for cell-specific tumor antigens [221,222]. A plug-and-play display strategy using ClyA fusions combined with the SpyTag/SpyCatcher technology was recently characterized as a versatile MV-based antigen display platform for tumor vaccination [223]. The strategy provided sufficient flexibility to efficiently load MVs with different tumor antigens, such as ovalbumin and tyrosinase-related protein 2 epitopes. This allows the simultaneous presentation of multiple, distinct tumor antigens to elicit a synergistic antitumor immune response and might be a promising step towards personalized therapy. In mouse models, subcutaneous injections of the tumor-antigen presenting MVs could abrogate lung melanoma metastasis and inhibit subcutaneous colorectal cancer growth. Via the Hbp autotransporter pathway, a papillomavirus tumor antigen, i.e., the human papillomavirus type 16 early protein E7, could be displayed on *E. coli* MVs, which induced E7 antigen-specific cellular immune responses and suppressed growth of grafted TC-1 tumors in mice upon subcutaneous immunization [224]. 

### 4.3. MVs Mediated Target Delivery 

The immunomodulatory potency, effective uptake by a variety of host cells, and general properties beneficial as an effector delivery system offers additional therapeutical applications for bacterial MVs in addition to their use as vaccine candidates. Recent studies suggest their exploitation along immunotherapy of cancer, including combinations with chemotherapy, gene therapy, and photothermal therapy. 

Remarkable anti-tumor efficacy was shown for MVs derived from *E. coli* ∆*msbB*, synthesizing underacylated, detoxified LPS. Intravenous injection of MVs into mice subcutaneously transplanted with CT26 murine colon adenocarcinoma resulted in MV accumulation in the tumor tissue and enhanced production of IFN-γ as well as IP-10, also known as CXCL10 [225]. MV treatment not only significantly reduced the tumor volume in a dose-dependent manner to complete the elimination of tumor tissue but also lowered the risk for lung metastasis. Similar results were obtained with MVs derived from *Lactobacillus acidophilus* MVs, indicating that diverse MVs derived from Gram-negative or -positive bacteria could exhibit anti-tumor activity [225]. While this seems a rather universal approach, other studies used more targeted approaches against specific features of tumors. Immune cell invasion or therapeutic agent penetration into tumor tissue can be blocked by excessive amounts of extracellular hyaluronic acid produced by tumors, in particular soft tissue sarcomas. Using the ClyA fusion strategy, MVs from a hypervesiculating *E. coli* Nissle strain were loaded with hyaluronidases and the cytotoxin ClyA [226]. Using a mouse tumor model, intravenous injection of the genetically engineered strain resulted in the accumulation of the MVs in subcutaneously located breast cancer cell tumors. The treatment remodeled the tumor stroma, most likely due to hyaluronic acid degradation combined with cytolytic activity, and ensured better accessibility of the tumor by chemotherapeutics, which increased their efficacy. Although detoxified versions of bacterial MVs have been reported, potential risks of adverse effects, in particular upon invasive administration, remain. Thus, a promising alternative might be synthetic bacterial outer membrane vesicles generated from *E. coli* outer membrane fractions using lysozyme and pH treatment [227]. The synthetic vesicles contained minor cytosolic contamination and induced low systemic pro-inflammatory responses compared to their natural OMV counterparts. Immunotherapeutic potential and adjuvant activity were confirmed in a mouse model upon co-administration of synthetic vesicles together with tumor tissue-derived vesicles, which induced a Th1- type T cell and humoral immune response. 

A recent study exploited the effective uptake of bacterial MVs by host cells and loaded MVs from attenuated *K. pneumonia* with the chemotherapeutic agent doxorubicin to generate chemotherapeutic agent nanocarriers [228]. Anti-tumor efficacy of such doxorubicin-loaded MVs was confirmed in cell culture as well as in vivo using a xenograft mouse model for lung cancer. Using biotechnological processes, such as extrusion and film-dispersion, MVs isolated from attenuated *Salmonella* spp. were decorated with a tumor-targeting tripeptide ligand using polyethylene glycol as a carrier and further coated with tegafur [229]. Systemic injection of these bioengineered MVs demonstrated effective inhibition of tumor growth, life-span extension in a melanoma mouse model, and reduced risk for melanoma development. 

Effector delivery via MVs can also be extended to RNA technology. Human epidermal growth factor 2 (HER2)-specific affibody decorated MVs from detoxified *E. coli* K-12 Δ*msbB* strains can effectively deliver siRNA to silence the expression of a kinesin spindle protein in cancer cells overexpressing HER2 [230]. Systemic injection of these MVs resulted in significant tumor regression in an animal model without notable side effects. A recent study introduced a rapid surface display technology for mRNA antigens on bacterial MVs [231]. Along the strategy, MVs are decorated with fusions of the RNA-binding protein L7Ae, the lysosomal escape protein listeriolysin O and ClyA. Such decorated MVs can rapidly absorb mRNA and deliver it into dendritic cells. In vivo studies show significant inhibition of melanoma progression and regression of colon cancer. In addition, immunized mice are protected from a tumor challenge. This “Plug-and-Display” strategy offers a personalized mRNA tumor vaccination based on MVs.

Photothermal therapy (PTT) belongs to one of the most recently introduced therapeutic strategies to treat tumors, with low toxicity and minimal invasiveness. PTT relies on agents converting light energy to heat upon exposure to a laser emitting near-infrared light, which kills the targeted cells. The combination of PTT with immunotherapy has demonstrated beneficial synergistic effects on primary tumor growth and prevention of metastasis [232]. In addition to the induction of anti-tumor immune responses upon intravenous injection of a single low dose of MVs from an attenuated *S. typhimurium*, extravasation of red blood cells was observed only in the tumor tissue, which allowed a precise and effective photothermal targeting of the tumors with a near-infrared laser [233]. *Salmonella* MVs have also been fused with tumor-derived melanoma cytomembrane vesicles and a near-infrared light-absorbing agent (poly (lactic-co-glycolic acid indocyanine green)) to generate hybrid vesicles, which integrate tumor-antigen presentation, self-adjuvant properties, and photothermal activity [234]. A similar approach fused *E. coli* MVs with B16-F10 cancer cell membranes and hollow polydopamine nanoparticles using ultrasonic treatment [235]. Anti-tumor efficacy of the hybrid vesicles was validated in a melanoma mouse model, demonstrating tumor-antigen presentation as well as accumulation of the vesicles in the tumor tissue allowing PTT. Another elegant study engineered a detoxified *E. coli* W3110 Δ*msbB* strain to overexpress the tyrosinase from *Rhizobium etli*, which is sufficient for the production of melanin [236]. Melanin not only accumulated in the bacterial cells but was also packaged into the MVs. Energy absorbance and heat production upon laser exposure was confirmed in vitro as well as in a 4T1 tumor-mouse model resulting in a reduction of tumor volume. 

While MVs released from bacteria can facilitate antibiotic resistance by the carriage of degradative enzymes or promoting horizontal gene transfer, they may also be used to deliver harmful antibiotics to bacterial populations. This approach seems tempting based on the cargo protection and potential delivery of concentrated doses to recipient cells mediated by MVs. Indeed, several reports highlight antimicrobial activity by a variety of naturally released MVs. The first evidence was provided for MVs derived from *P. aeruginosa* MVs exhibiting antimicrobial activity against other bacteria [36,65]. Subsequent studies confirmed the antimicrobial activity of MVs derived from various Gram-negative bacteria and correlated with the presence of autolysins in the MVs [237,238]. MVs from *Chromobacterium violaceum* carry violacein, a purple hydrophobic bisindole, which has antibiotic activity predominantly against Gram-positive bacteria, such as *S. aureus* [239]. Bacterial MVs have also demonstrated efficacy against biofilm communities and may facilitate their dispersion. For example, MVs from *Lacticaseibacillus casei* have demonstrated anti-biofilm activity against *S. enterica* serovar Enteritidis and *Burkholderia thailandensis* MVs against methicillin-resistant *S. aureus* and *Streptococcus mutants* biofilms [240,241,242]. Anti-biofilm activity was attributed to peptidoglycan hydrolases present in *L. casei* and *B. thailandensis* MVs as well as 4-hydroxy-3-methyl-2-(2-non-enyl)-quinoline and long-chain rhamnolipids being the cargo of MVs from *B. thailandensis.* Notably, MV-treated biofilms showed altered morphology and higher susceptibility to antibiotic treatment [240,242]. This might offer combined therapeutic approaches with conventional antibiotics, which alone have generally low efficacy against bacteria protected in biofilms. 

Micro-predators such as *Myxococcus xanthus* and *Lysobacter enzymogenes* release MVs containing a cocktail of bioactive compounds and degradative enzymes, including sugar hydrolases, monooxygenases, and proteases, to kill their prey such as other bacteria, fungi, and oomycetes [243,244,245,246]. Thus, comprehensive characterization of the cargo might unravel new anti-bacterial and anti-fungal compounds. 

Bacterial MVs produced during antibiotic exposure may incorporate the attacking agents described for gentamycin-packed MVs released from *P. aeruginosa* during exposure to the antibiotic [65]. Notably, these gentamycin-carrying MVs showed even higher bacteriolytic activity against Gram-negative and -positive bacteria than natural *P. aeruginosa* MVS. These initial observations were extended by a study loading *A. baumannii* MVs with a variety of antibiotics, i.e., ceftriaxone, amikacin, azithromycin, and levofloxacin, which not only effectively killed a variety of bacterial pathogens in vitro but also reduced ETEC colonization levels upon oral treatment of infected mice [247]. Additionally, co-incubation with bacteria during MV production, sonication, electroporation, or extrusion of purified MVs have been discussed in the literature as potential loading techniques [248,249]. However, efficient protocols for such approaches still remain to be established.

**Table 1 antibiotics-12-01045-t001:** Representative studies characterizing diverse therapeutic opportunities of MVs.

Bacterial MV Donor	Heterologous Effectors Included	(Potential) Therapeutic Application	Experimental Validation	Ref. ^i^
*Acinetobacter baumannii*	-	vaccine candidate against donor bacterium	humoral, protective immune response against pneumonia, and sepsis [m] ^ii^	[170,171]
*A. baumannii*	ceftriaxone, amikacin, azithromycin and levofloxacin	anti-microbial therapy	humoral, protective immune response against pneumonia, and sepsis [m]	[247]
*Bacteriodes fragilis*	-	treatment against colitis	prevents experimental colitis symptoms [m]	[157]
*Bordetella pertussis*	-	vaccine candidate against donor bacterium	humoral immune response [m]	[182]
*Burkholderia thailandensis*	-	anti-microbial therapy	anti-microbial and anti-biofilm efficacy against MRSA and *Streptococcus mutans*	[241,242]
*Chromobacterium violaceum*	-	anti-microbial therapy	anti-microbial activity mainly against Gram-positive bacteria due to violacein	[239]
*Escherichia coli*	H1-type haemagglutinin from influenza A virus (H1N1) and RBD from MERS-CoV	vaccine candidate against H1N1pdm09 and MERS-CoV	humoral, neutralizing immune response against both viruses, and protection against H1N1 influenza virus [m]	[217]
*E. coli*	human papillomavirus early E7 protein from via Hbp fusion	anti-tumor therapy	specific cellular immune response against TC-1 tumors [m]	[224]
*E. coli*	KMP-11 and Linf08.1190 from *Leishmania donovanii* via AIDA fusion	vaccine candidates against leishmaniasis	humoral immune response [m]	[210]
*E. coli*	ovalbumin and TRP2 epitopes via ClyA fusions and SpyTag/SpyCatcher system	anti-tumor vaccination and therapy	reduction of tumor growth and prevention of metastasis [m]	[223]
*E. coli*	Omp22 of *A. baumannii* via ClyA fusion	vaccine candidate against *A. baumannii*	humoral, protective immune response [m]	[208]
*E. coli*	RBD from SARS-Cov-2 via ClyA fusion	vaccine candidate against SARS-CoV-2 infection	humoral immune response [m]	[213]
*E. coli*	tumor tissue-derived vesicles fused with bacterial MVs	anti-tumor therapy	humoral and cellular immune response [m]	[227]
*E. coli* Δ*lpxM*	stEsxA, stSbi and stSpA *Staphylococcus aureus* via the SpyTag/SpyCatcher system	vaccine candidate against *S. aureus*	humoral, protective immune response [m]	[212]
*E. coli* Δ*msbB* and *Lactobacillus acidophilus*	-	anti-tumor therapy	anti-tumor cytokine response and reduction of CT26 tumors [m]	[225]
*E. coli* Δ*msbB*	siRNA-HER2 and Her2-affibody	anti-tumor therapy	reduction of tumor size [m]	[230]
*E. coli* Δ*msbB*	melanin	anti-tumor photothermal therapy	accumulation in 4T1 tumor tissue and effective photothermal therapy [m]	[236]
*E. coli* Δ*msbB*	RNA-binding-protein L7Ae, listerolysin O and ClyA	personalized mRNA-based anti-tumor vaccine candidate	inhibition of melanoma progression and regression of colon cancer [m]	[231]
*E. coli* Δ*tolRA*	O-antigen of *Francisella tularensis*	vaccine candidate against *F. tularensis*	humoral, protective immune response [m] ^i^	[205]
*E. coli* Δ*tolRA*	ESAT6, Ag85B, and Rv2660v from *Mycobacterium tuberculosis* via fusions to Hbp	vaccine candidate against *M. tuberculosis*	detection on vesicle surface	[209]
*E. coli* Δ*wecA*	K1 and K2 polysacharides from *Klebsiella pneumoniae*	vaccine candidate against *K. pneumoniae* infection	humoral, protective immune response [m]	[206]
*E. coli* Nissle 1917	-	treatment against colitis	Protection against DSS-induced colitis [m]	[155]
*E. coli* Nissle 1917 Δ*nlpI*	Hyaluronidase via ClyA fusion	anti-tumor therapy	Stroma modulation in tumor tissue enhancing efficacy of immunotherapy [m]	[226]
non-typable *Haemophilus influenzae* (NTHi)		treatment against colitis	humoral, protective immune response against heterologous NTHi strains [m]	[185]
enterotoxigenic *E. coli* (ETEC) Δ*msbB* Δ*eltA* and *Vibrio cholerae* Δ*msbB* Δ*ctxAB*	-	combined vaccine candidate against cholera and ETEC	humoral, protective immune response against both pathogens [m]	[190]
ETEC Δ*msbB* Δ*eltA* Δ*ompA* and *V. cholerae* Δ*msbB* Δ*ctxAB* Δ*ompA*	RBD from SARS-CoV-2 via Lpp-OmpA fusion	combined vaccine candidate against SARS-CoV-2 and donor bacteria	humoral, neutralizing immune response against SARS-CoV-2 [m]	[200]
*K. pneumonia*	Doxorubicin	anti-tumor therapy	growth inhibition of A549 tumors [m]	[228]
*Lacticaseibacillus casei*	-	anti-microbial therapy	anti-biofilm activity against *Salmonella enterica* sv. Enteritidis	[240]
*Lactobacillus* ssp. (*i.e., L. paracasei, L. plantarum Q7, L rhamnosus*)	-	treatment against colitis	protection against DSS-induced colitis [m]	[158,159,160]
*L. plantarum*	fucoxanthin	treatment against colitis	improved protection against DSS-induced colitis [m]	[162]
*Lactococcus lactis*	hepatitis C core antigen	vaccine candidate against hepatitis C virus	humoral immune response against the hepatitis C virus [m]	[219]
*Lysobacter enzymogenes*	-	anti-fungal therapy	anti-fungal activity due to HSAF against various fungi [iv] ^iii^	[246]
*M. tuberculosis*	-	vaccine candidate against donor bacterium	humoral, protective immune response [m]	[173]
*Myxococcus xanthus*	-	anti-microbial therapy	anti-microbial activity against a variety of Gram-negative bacteria [iv]	[243,244]
*Neisseria meningitids*	fHbp, NadA and NHBA	diverse vaccine candidates (e.g., MenBvac^®^ MenZB^®^, VA-MENGOC-BC^®^, and Bexsero^®^) against meningococcal disease	humoral, protective immune response [m], [h]	[144,145,146,147,164]
*N. meningitidis*	recombinant RBD from SARS-CoV-2 mixed with Bexsero^®^	combined vaccine candidate against meningococcal disease and SARS-CoV-2	humoral immune response against the spike protein [m]	[215]
*N. meningitids*	recombinant fusion protein (truncated core and NS3 from hepatitis C virus) mixed with MVs	vaccine candidate against hepatitis C virus	humoral immune response against the hepatitis C virus [m]	[220]
*N. meningitids* Δ*lpxL2* Δ*synX*	OpcA, fHbp, PorA	vaccine candidates against donor bacterium	humoral, protective immune response [m], [h] ^v^	[193,194]
*N. meningitidis* Δ*porA* Δ*siaD* Δ*lpxL1* Δ*rmpM*	Spike protein from SARS-CoV-2 via mCramp fusion	vaccine candidate against SARS-CoV-2	humoral, protective immune response against SARS-CoV-2 [sh] ^iv^, [m]	[214]
*Pseudomonas aeruginosa*	-	anti-microbial therapy	anti-microbial activity against variety of bacteria [iv]	[36,65,237]
*Salmonella enterica* sv. Typhimurium Δ*tolRA*	MOMP of *Chlamydia trachomatis* via Hbp fusion	vaccine candidates against *C. trachomatis*	detection on vesicle surface	[209]
*S. enterica* sv. Typhimurim Δ*tolRA* Δ*msbB*	RBD from SARS-Cov-2 via the SpyTag/SpyCatcher system	vaccine candidate against SARS-Cov-2	humoral, protective immune response against SARS-Cov-2 [sh]	[216]
*Salmonella* spp.	melanoma cytomembrane vesicles and poly(lactic-co-glycolic acid indocyanine green fused with bacterial MVs	anti-tumor photothermal therapy	effective photothermal therapy and anti-tumor immune response [m]	[234]
*Salmonella* spp.	tumor-targeting tripeptide ligand and tegafur	anti-tumor therapy	inhibition tumor growth and life-span extension [m]	[229]
*Salmonella* spp. ΔppGpp	-	anti-tumor photothermal therapy	MV-induced extravasation enhances efficacy of photothermal therapy [m]	[233]
*Shigella sonnei* Δ*tolR::kan* Δ*virG::nadAB* Δ*htrB::cat*	-	vaccine candidate (1790GAHB) against donor bacterium	humoral, protective immune response [m], [h]	[192,195,196,197,198]
*Shigella* spp.	-	vaccine candidate against donor bacterium	humoral, protective immune response against shigellosis [m]	[188]
*S. aureus*	-	vaccine candidate against donor bacterium	humoral, protective immune response against lethal dose challenge [m]	[172]
*S. aureus* Δ*agr*	EDIIIconA and EDIIIconB from dengue virus via fusion to *S. aureus* carrier proteins	vaccine candidate against dengue virus	humoral, protective immune response against all four dengue virus serotypes [m]	[218]
*Streptococcus pneumoniae*	-	vaccine candidate against donor bacterium	humoral, protective immune response [m]	[174,175]

^i^ Ref. = References; ^ii^ [m] = demonstrated in mouse model; ^iii^ [iv] = demonstrated in vitro; ^iv^ [sh] = demonstrated in Syrian hamster model; ^v^ [h] = demonstrated in human clinical trial.

## 5. Conclusions and Future Directions

It is becoming increasingly evident that all bacteria can release MVs, which may originate via different biogenesis routes. In context with the human body, the gastrointestinal tract harbors a diverse microbial community with 100 trillion members, which can reach a biomass of 1.5 kg [250]. As presented in this review, several studies indicate that in vivo conditions can facilitate MV release suggesting an enormous amount of MVs produced by the microbes residing in the human gastrointestinal tract. Herein, we focus on MVs derived from bacterial pathogens and highlight their beneficial impact on bacterial fitness and survival during in vivo colonization. Recent findings suggest a vicious cycle of MV biogenesis, pathophysiological function, and antibiotic therapy. Proof-of-principle studies indicate that host defense mechanisms and antimicrobial therapy facilitate MV release, which counteracts the treatment and can exacerbate the disease symptoms. Moreover, MVs can confer local antibiotic resistance via the carriage of degradative enzymes and aid in the emergence of multidrug-resistant strains. The release mechanism, growth condition, and donor species affect the composition of these multifactorial complexes, exponentially increasing their potential activities. Despite significant advances in MV characterization and unraveling a vast array of functions, we are just beginning to understand all features associated with MVs. Thus, future studies on MVs will likely provide novel insights into deciphering their contribution to bacterial physiology and host modulation. 

Several studies also highlight the potential of MVs for future medical applications, such as antimicrobial therapy, vaccination, and targeted delivery. However, among all the proposed applications, only OMV-based vaccines against *N. meningitidis* have been approved and are commercially available. Challenges remain in upscaling production, purification, quality control, and safety. Techniques identifying and separating the heterogenous MV types produced within a bacterial culture are currently lacking. Additional concerns are undesired changes in the MV composition and the uncontrollable systemic spread of MVs. Aside from a better understanding of the versatile interactions of MVs with host cells, defined and standardized protocols and methods for quantity and quality control as well as reactogenicity and toxicity assays, are required. These issues must be addressed in future research initiatives. In the era of multi-drug resistant pathogens, we should take advantage of MVs and invest in the development, approval, and commercialization of MV-based therapeutic applications. 

## Figures and Tables

**Figure 1 antibiotics-12-01045-f001:**
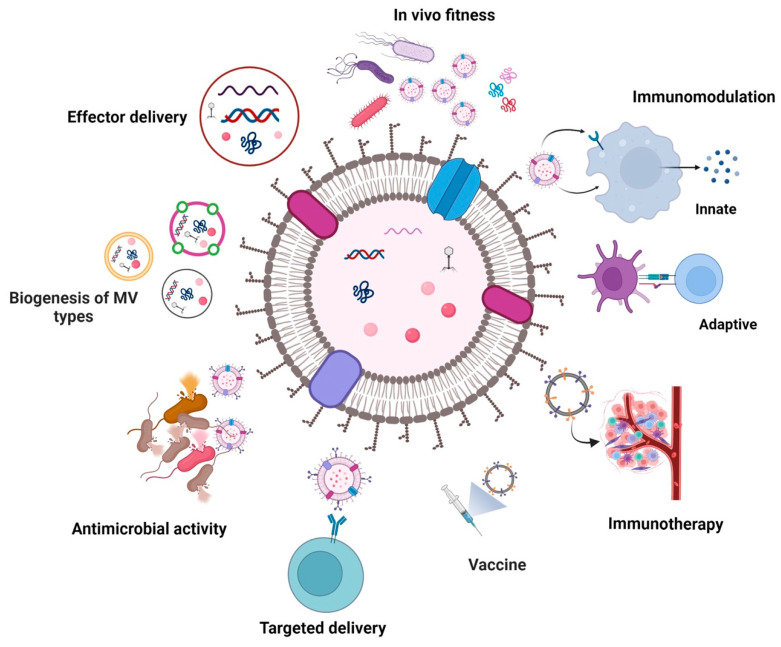
Graphical overview of the MV-related topics addressed in this review. MVs can originate from Gram-negative and -postive bacteria via different biogenesis routes affecting their composition. Although the figure depicts a single bilayered MV, the schematic graphic should represent all types of bacterial MVs known. Hence it comprises single and double-bilayered MVs, which include blebbing type MVs (such as outer membrane vesicles, cytoplasmic membrane vesicles, and outer-inner membrane vesicles) and MVs originating from explosive cell lysis (such as explosive outer membrane vesicles, explosive cytoplasmic membrane vesicles, and explosive outer-inner membrane vesicles). Although energetically costly, MVs facilitate bacterial adaptation and survival. In particular, MVs derived from human pathogens have been reported to mediate effector delivery, promote in vivo fitness and modulate immune responses of the host. On the other hand, unique features of MVs can be exploited to utilize them for therapeutic applications, such as targeted delivery of bioactive compounds, vaccination, immunotherapy, or antimicrobial treatment. Created with BioRender.com (accessed on 30 May 2023).

**Figure 2 antibiotics-12-01045-f002:**
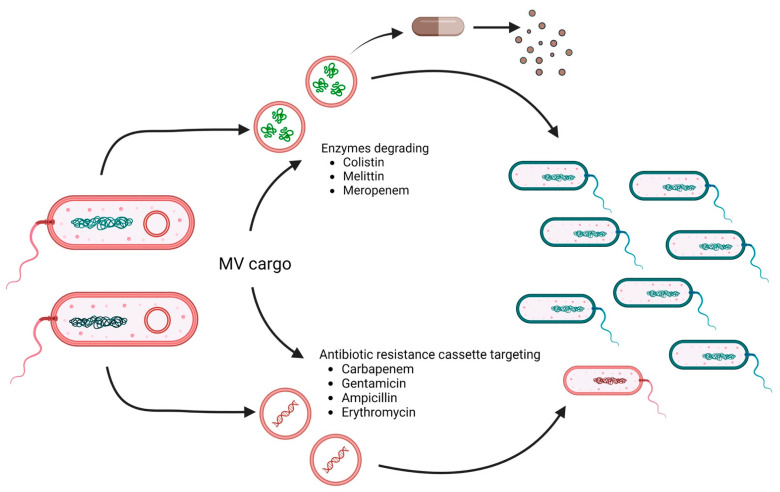
MV cargo can facilitate antimicrobial resistance. MVs derived from resistant bacteria (red) can carry antibiotic-degrading enzymes and/or DNA encoding antibiotic-resistance genes. Thus, MV cargo can either degrade antimicrobial agents (upper pathway) to confer temporal protection survival of sensitive bacteria (blue) or promote horizontal gene transfer (lower pathway), resulting in the emergence of new resistant bacteria (red). Relevant antibiotics reported to be affected by the MV cargo are listed. Created with BioRender.com (accessed on 30 May 2023).

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
