# Peer review of "The Two Faces of Bacterial Membrane Vesicles: Pathophysiological Roles and Therapeutic Opportunities"

_antibiotics, 2023, doi:10.3390/antibiotics12061045_

Round 1

Reviewer 1 Report

The authors reviewed the role of the bacterial membrane vesicles in bacterial adaptation, its biosynthesis, and its role in the development of therapeutic approaches. Many aspects of bacterial membrane vesicles were recently reviewed in excellent review articles, such as their roles in bacterial adaptation (10.1007/s00203-014-1042-7), their composition (10.1038/s41579-023-00875-5), their biosynthesis (10.1038/s41579-018-0112-2), their application in vaccines development (10.1016/j.addr.2022.114294, 10.20892/j.issn.2095-3941.2022.0452), in drug delivery (10.3390/pharmaceutics15041052 10.3390/pharmaceutics15041052), and therapeutics applications (10.3390/ijms18061287). Although the manuscript is very informative and well-structured, I believe that the authors must describe clearly what are the innovative and unique points of their article (if there are any). This aspect is not clear and produced a negative impact on the manuscript's originality. I believe that only after this point could be informed and discussed the manuscript will be suitable for publication in a high-impact journal.

Other points that the authors are invited to improve their manuscript are regarding the absence of methodological information about the keywords, time of the search, and databases used in their search for articles. This is an important point that gives reproducibility to the review article.

Author Response

The authors reviewed the role of the bacterial membrane vesicles in bacterial adaptation, its biosynthesis, and its role in the development of therapeutic approaches. Many aspects of bacterial membrane vesicles were recently reviewed in excellent review articles, such as their roles in bacterial adaptation (10.1007/s00203-014-1042-7), their composition (10.1038/s41579-023-00875-5), their biosynthesis (10.1038/s41579-018-0112-2), their application in vaccines development (10.1016/j.addr.2022.114294, 10.20892/j.issn.2095-3941.2022.0452), in drug delivery (10.3390/pharmaceutics15041052 10.3390/pharmaceutics15041052), and therapeutics applications (10.3390/ijms18061287). Although the manuscript is very informative and well-structured, I believe that the authors must describe clearly what are the innovative and unique points of their article (if there are any). This aspect is not clear and produced a negative impact on the manuscript's originality. I believe that only after this point could be informed and discussed the manuscript will be suitable for publication in a high-impact journal.

Author response:

We thank the reviewer for constructive criticism on our manuscript and for providing helpful comments that helped us to further improve this review article.

We also thank the reviewer for highlighting that our review is very informative and well-structured.

We agree with the reviewer that several excellent reviews on bacterial MVs have been published recently. That is why, we specifically focused in this review on bacterial pathogens and emphasize recent findings suggesting a vicious cycle of MV biogenesis, pathophysiological function and antibiotic therapy. This link has never been addressed in a review so far and therefore represents a novel aspect presented herein, in particular in chapter 2 and 3. Chapter 4 provides a comprehensive overview of therapeutical applications ranging from immunomodulation, MV-based vaccines and targeted delivery. Although individual parts of these topics have been addressed in other reviews, they have not been presented in a combined fashion recently. Importantly, only such a comprehensive overview of the diverse therapeutical applications allows a reader to identify paralleled strategies and outcomes. For example, the use of similar decoration/ loading strategies along vaccine studies and effector delivery. As mentioned by the reviewer, the last review article comprehensively covering all therapeutical applications was published in 2017 (10.3390/ijms18061287). As our review also highlights new, primary research of the last six years we think the current review article provides sufficient novel information in a combined format.

In order to emphasize the above mentioned vicious cycle along the pathophysiological function as well as the comprehensive overview of therapeutical applications more clearly, we re-phrased the abstract in the revised version.

Overall, we hope that we could clarify that the current review bears sufficient originality.

Other points that the authors are invited to improve their manuscript are regarding the absence of methodological information about the keywords, time of the search, and databases used in their search for articles. This is an important point that gives reproducibility to the review article.

Author response: We find the reviewer’s idea supportive and added this information at the end of the introduction.

We state:

“To assemble the relevant studies covered in this review a comprehensive literature search was performed in the PubMed or Google Scholar database for original, peer-reviewed research publications. No restriction on publication date was applied. Search terms comprised the term bacterial membrane vesicle in combination with keywords listed for this review, appropriate phrases found in the abstract or titles of individual chapters. The bibliographies of relevant articles were also searched to identify other related studies.“

Reviewer 2 Report

Emergence of antimicrobial resistance is growing concern for global healthcare industry and medicine. Authors have discussed, recent developments in diverse MV biogenesis routes and explain physiological functions of MVs derived from human pathogens with respect to its adaptation and cellular fitness.

The manuscript could be considered for publication after careful revision.

Some comments:

1.       I recommend the authors to provide statistical data in abstract and conclusion.

2.       How antimicrobial resistance is threatening the global healthcare, needs to be addressed in discussion.

3.       The figure 1 is excellently drawn, authors should explain whether the MVs are single layered or double layered, please correct accordingly.

4.       Imaging of bacterial MVs would a good option to visualized resistant microorganisms. I suggest relevant update on imaging of MVs in live cell. For example, Kadam, U.S., Lossie, A.C., Schulz, B., Irudayaraj, J. (2013). Gene Expression Analysis Using Conventional and Imaging Methods. In: Erdmann, V., Barciszewski, J. (eds) DNA and RNA Nanobiotechnologies in Medicine: Diagnosis and Treatment of Diseases. RNA Technologies. Springer, Berlin, Heidelberg. https://doi.org/10.1007/978-3-642-36853-0_6

5.       Can authors provide list of antibiotics, against which resistance is developed and what measures are taken to overcome such resistance.

6.       Author could include more figures and schemes to increase appeal of the manuscript.

Language is acceptable.

Author Response

Reviewer 2:

Emergence of antimicrobial resistance is growing concern for global healthcare industry and medicine. Authors have discussed, recent developments in diverse MV biogenesis routes and explain physiological functions of MVs derived from human pathogens with respect to its adaptation and cellular fitness.

The manuscript could be considered for publication after careful revision.

Author response:

We thank the reviewer for constructive criticism on our manuscript and for providing helpful comments that helped us to further improve this review article.

Some comments:

  1. I recommend the authors to provide statistical data in abstract and conclusion.

Author response: We agree with the reviewer. The revised version contains more data on antimicrobial resistance cases incl. burden for the global health care as well as on microbial biomass in the human body, which is capable of producing MVs. However, in our opinion the abstract with a strict word limit (200 words maximum) seems not the appropriate place to put this information. Thus, we have added this data into the introduction and discussion.

  1. How antimicrobial resistance is threatening the global healthcare, needs to be addressed in discussion.

Author response: We agree with the reviewer and added more information on antimicrobial resistance cases and burden for the global health care systems in the discussion.

  1. The figure 1 is excellently drawn, authors should explain whether the MVs are single layered or double layered, please correct accordingly.

Author response: Although the figure depicts a single bilayered MV, the schematic graphic should represent all types of bacterial MVs known. Hence it comprises single and double bilayered MVs, which include blebbing type MVs (such as outer membrane vesicles, cytoplasmic membrane vesicles and outer-inner membrane vesicles) and MVs originating from explosive cell lysis (such as explosive outer membrane vesicles, explosive cytoplasmic membrane vesicles and explosive outer-inner membrane vesicles). We have added a statement in the figure legend for clarification.

  1. Imaging of bacterial MVs would a good option to visualized resistant microorganisms. I suggest relevant update on imaging of MVs in live cell. For example, Kadam, U.S., Lossie, A.C., Schulz, B., Irudayaraj, J. (2013). Gene Expression Analysis Using Conventional and Imaging Methods. In: Erdmann, V., Barciszewski, J. (eds) DNA and RNA Nanobiotechnologies in Medicine: Diagnosis and Treatment of Diseases. RNA Technologies. Springer, Berlin, Heidelberg. https://doi.org/10.1007/978-3-642-36853-0_6

Author response: We added a statement at the end of chapter 3.1. that the ongoing development of improved high-resolution imaging technologies may allow a better detection and identification of MVs harboring resistance markers. We cite the mentioned literature accordingly.

  1. Can authors provide list of antibiotics, against which resistance is developed and what measures are taken to overcome such resistance.

Author response: We agree with the reviewer. Please see below our response to comment 6 for details.

  1. Author could include more figures and schemes to increase appeal of the manuscript.

Author response: We agree with comments 5 & 6 of the reviewer and provided a new Figure 2 which graphically summarizes the antibiotic resistance emergence via MVs and lists the relevant antimicrobial agents/ resistance genes. With regard to the measures taken to overcome MV-based resistance no applicable strategies are currently available, however we added a thought on a potential therapeutic intervention.

We state:

„Moreover, therapeutic agents reducing or blocking MV release might be available in the future and could be valuable tools to counteract such MV-based antimicrobial resistance mechanisms [93].”

Reviewer 3 Report

Title: The two faces of bacterial membrane vesicles: Pathophysiological roles and therapeutic opportunities

This manuscript is not ready to publish in the journal as many weak points were presented in it. However, I do believe that if they can improve the manuscripts following all comments. It might have a chance to publish in the journal.

Comments

1. Topic: The title is not clear and should be modified.

2. Line 9-11: please remove or re-write.

3. The first sentence of the abstract should describe the definition of bacterial membrane vesicle, followed by the importance of the membrane vesicle.

4. Line 20: I found a something error in the letters. Please check.

5. Line 22: in vivo should be written in italic.

6. Please identify the objective of this study. It may be better if the author uses passive voice instead of, we discuss ……

7. Line 26: Five Keywords are maximum. Please remove unimportant keywords.

8. Line 59 and Figure 1: in vivo should be written in italic.

9. Line 85-87: please re-write

10. The authors describe the importance of membrane vesicle by description. It is better if the authors summarize the information in Table or Figure. Then, the authors explain the data from the tables.

11. Some headings, the authors explain the data like literature review. Please summarize the key information. It is better.

12. The information in the text is not related to the abstract. Please re-write the abstract.

13. There are many references. Please delete some unnecessary references.

14. The references of 2022,2021 (and 2023) are suggested to be cited.

This manuscript is well-written by the authors.

Author Response

Reviewer 3:

Title: The two faces of bacterial membrane vesicles: Pathophysiological roles and therapeutic opportunities

This manuscript is not ready to publish in the journal as many weak points were presented in it. However, I do believe that if they can improve the manuscripts following all comments. It might have a chance to publish in the journal.

Author response:

We thank the reviewer for constructive criticism on our manuscript and for providing helpful comments that helped us to further improve this review article.

Comments

  1. Topic: The title is not clear and should be modified.

Author response: We are open for alternative suggestions of a better suited title. However, we believe that the current title of this review article is appealing and will attract a broad readership. We also think that the covers the most important aspects discussed in this review article. If you are very critical, the only part not reflected by the title is the “biogenesis” section. This chapter is essential to understand the link between MV biogenesis, pathophysiological function and antibiotic therapy, but not necessarily the focus of the review. In summary, we still think that the current title is a good eye-catcher and the content and outline of the review are now clearly stated in the revised abstract. However, we encourage the reviewer to propose a better suited title, if she/ he still believes that the current title should be changed.

  1. Line 9-11: please remove or re-write.

Author response: We agree and removed this statement from the abstract.

  1. The first sentence of the abstract should describe the definition of bacterial membrane vesicle, followed by the importance of the membrane vesicle.

Author response: We agree and re-phrased the abstract.

  1. Line 20: I found a something error in the letters. Please check.

Author response: Although not specified, we think the reviewer pointed out a missing “the”. Thus, we corrected the phrase “…we discuss current…” to  “…we discuss the current…”

  1. Line 22: in vivo should be written in italic.

Author response: We agree and italicized “in vivo” throughout the text.

  1. Please identify the objective of this study. It may be better if the author uses passive voice instead of, we discuss ……

Author response: This comment might arise from a misunderstanding. We would kindly emphasize, that the manuscript is an “invited review article” and not a primary research article. Thus, our article rather covers published literature and not follows an empirical validation of a hypothesis. We think that the revised abstract and the last paragraph of the introduction concisely summarize the content of the review and the novel aspects addressed in the review article. As the manuscript is an “invited review article”, the use of active voice “we discuss …” seems appropriate.

  1. Line 26: Five Keywords are maximum. Please remove unimportant keywords.

Author response: We respectfully disagree. The author guidelines on the journal website (https://www.mdpi.com/journal/antibiotics/instructions) clearly specifies that up to ten keywords can be added: The website states: “Keywords: Three to ten pertinent keywords need to be added after the abstract.”

  1. Line 59 and Figure 1: in vivo should be written in italic.

Author response: We agree and italicized “in vivo” throughout the text.

  1. Line 85-87: please re-write

Author response: We agree and re-phrased the sentence. It now reads:

“Since first reports of Escherichia coli blebs almost 60 years ago [23], several in-depth studies focused on the biogenesis of vesicles derived from Gram-negative and Gram-positive bacteria [11,24].“

  1. The authors describe the importance of membrane vesicle by description. It is better if the authors summarize the information in Table or Figure. Then, the authors explain the data from the tables.

Author response: We would kindly point the reviewer to Figure 1, which provides a summary of the important MV features discussed in the review article and therefore exactly provides the item requested by the reviewer.

  1. Some headings, the authors explain the data like literature review. Please summarize the key information. It is better.

Author response: This comment might arise from a misunderstanding. We would kindly emphasize, that the manuscript is an “invited review article” and not a primary research article. Thus, titles explaining the content like in a literature review is quite appropriate for this type of article.

  1. The information in the text is not related to the abstract. Please re-write the abstract.

Author response: We agree that the entry of the abstract might have been beyond the scope of the review article. Thus, we re-phrased the entire abstract, which now summarizes the content of the review and highlights the novel aspects addressed in the review article.

  1. There are many references. Please delete some unnecessary references.

Author response: We respectfully disagree. For a review article the amount of references is appropriate. We would like to point out that several reviews published recently in “Antibiotics” contain a similar amount of references.

E.g.:

  • Antimicrobial Resistance: Two-Component Regulatory Systems and Multidrug Efflux Pumps. https://doi.org/10.3390/antibiotics12060965

Number of citations: 376

  • Clinical Impact of Staphylococcus aureus Skin and Soft Tissue Infection. https://doi.org/10.3390/antibiotics12030557

Number of citations: 248

  • Evolutionary Dynamics between Phages and Bacteria as a Possible Approach for Designing Effective Phage Therapies against Antibiotic-Resistant Bacteria. https://doi.org/10.3390/antibiotics11070915

Number of citations: 224

  • The Use of Bacteriophages in Biotechnology and Recent Insight into Proteomics. https://doi.org/10.3390/antibiotics11050653

Number of citations: 213

  • Polymicrobial Infections and Biofilms: Clinical Significance and Eradication Strategies. https://doi.org/10.3390/antibiotics11121731

Number of citations: 207

  1. The references of 2022,2021 (and 2023) are suggested to be cited.

Author response: We agree with the reviewer that a review article should cite recent, timely research articles. Nonetheless, it is appropriate to also cite older literature for the historical perspective. As explained in the response to comment 13, the amount of references seems appropriate and is in a similar range to previously published review articles in “Antibiotics”.

Round 2

Reviewer 1 Report

Thank you for having given me the opportunity to review again the manuscript entitled “The two faces of bacterial membrane vesicles: Pathophysiological roles and therapeutic opportunities”. Despite the existence of some overlapping with published reviews, I believe that the manuscript is attractive and informative and represents a contribution to the field. I am satisfied with the answers of the authors.  Thus, I believe that the manuscript meets the requirements for publication in Antibiotics, and I recommend accept it in its current form.